# A Non-Liner Decision Model for Green Crowdfunding Project Success: Evidence from China

**DOI:** 10.3390/ijerph16020187

**Published:** 2019-01-10

**Authors:** Jinbi Yang, Libo Liu, Chunxiao Yin

**Affiliations:** 1School of Business, Jiangnan University, Wuxi 214122, China; yangjinbi@jiangnan.edu.cn; 2Department of Business, Technology and Entrepreneurship, Swinburne University of Technology, Hawthorn, Victoria 3122, Australia; 3College of Computer & Information Science, Southwest University, Chongqing 400715, China; yincx@swu.edu.cn

**Keywords:** environmental sustainability, green crowdfunding project, decision making model, goal setting, project duration, non-linear

## Abstract

Green growth and environmental sustainability have become a significant focus in today’s living. We believe that green crowdfunding project can make an important contribution to the creation and evaluation of environmental systems. This study aims to investigate the determinants of green crowdfunding project success. Contrary to the linear relationship in previous literature, we propose a non-liner decision model that includes three determinants, funds pledge, goal setting, and project duration to predict project success. The quantitative approach method was employed. We crawl data on 1389 green crowdfunding projects from Tencent Lejuan, a crowdfunding platform in China. By using ordinary least square method to conduct data analysis, we find that the effect of goal setting on project success is non-linear as low and moderate levels of goal setting are not always likely to have a significant impact on project success, but the presence of a higher goal is likely to exert a positive effect on project success. Moreover, results show that the effect of project duration on project success is non-linear as short and moderate levels of duration is not always certain to have significant impact on project success, but the presence of a long duration is likely to exert a positive effect on project success. This study has implications for fund-seekers for green crowdfunding projects and managers of crowdfunding platforms.

## 1. Introduction

Environment pollution has become increasingly critical for people. As Internet has promoted the development of the business and society, “Internet + Environment Protection” should attract more attention. Crowdfunding platforms focusing on environment protection take advantage of the Internet to extend the boundary of the channels for public welfare activities related to the environment [1,2]. These platforms launch environmental projects, including tree planting and sand prevention, marine protection, garbage classification, clean water sources and so on. In such way, individuals could participate in environmental groups and engage in public welfare activities online rather than offline. The first low-carbon crowdfunding project based on the Chinese certified emission reduction (CCER) was delivered in Hubei Province of China in 2015, promoting low-carbon economic transformation and curb greenhouse gas emissions [3]. A famous Chinese green crowdfunding platform “Drive to Green” has hosted a successful crowdfunding project for “activities of agreement of one cup of water” since 2016, which helped nearly 3000 teachers and students in seven schools of Gansu Province and Anhui Province in China province to get clean water [4]. Green crowdfunding project can make an important contribution to the creation and evaluation of environmental systems, which break new ground in environmental responsibility and ultimately improve the natural environment [5]. 

*Green crowdfunding project*, including green donation-based and green reward-based crowdfunding project, takes advantage of the Internet to extend the boundary of the channels of environmental public welfare activities [6]. For green donation-based crowdfunding platform, some environmental projects are launched by some nonprofit organizations, such as tree planting and sand prevention, marine protection, garbage classification, and clean water sources. Our work focuses on green donation-based crowdfunding. The most famous and largest donation-based platform, CrowdRise, raised more than 1.5 million dollars for donation-based projects in 2015 [7]. Green donation-based crowdfunding also has been gaining popularity in China in the past several years. In 2015, a professional green donation-based crowdfunding platform was established in China, named Drive to Green [8]. More importantly, the emergence of various environment protection crowdfunding platforms encourages individuals and businesses to fulfill their social responsibility of environmental protection significantly. Hence, the current research aims to explore how a green donation-based crowdfunding project can success. 

Existing studies have considered the effects of design strategies (i.e., reward schema), project quality (i.e., high quality of project description), social networks, and social capital on crowdfunding project success from the perspective of capital seekers or fundraisers [9,10,11,12,13]. However, for green crowdfunding projects, most of the fundraisers are non-profit environmental organizations; hence, conclusions related to reward schema, social networks, and social capital may be directly applicable. Fundraisers of green crowdfunding projects can properly set their projects to gain support from the public. Goal and duration are two common questions that should be decided by fundraisers when launching crowdfunding projects [14,15,16,17]. Empirical results revealed inconsistent findings about the effect of goal setting on project success that some research indicated negative influences of goal setting [11,12,17,18,19,20,21] and some indicated no effects [11] or positive effects [18]. Similar inconsistent findings were found for the effect of project duration that positive effects [17,19,20], negative effects [12,18], and no effects [11,21] were all identified. These conflicting accounts suggest that the relationships between goal setting, duration and project success are more complex than has been anticipated by previous empirical studies. To reconcile inconsistent findings, this study propose a non-liner decision model and investigate the non-linear relationship of project setting (goal setting and project duration) on project success in the green crowdfunding context.

We collected data on a leading crowdfunding platform in China: Tencent Lejuan crowdfunding platform (https://gongyi.qq.com/succor/index.htm). We focus on green crowdfunding projects; thus, we only collect data from projects related to green growth in Tencent Lejuan crowdfunding platform. Our results show that the effect of goal setting on project success is non-linear as low and moderate levels of goal setting are not always likely to have significant impact on project success, but the presence of a higher goal is likely to exert a positive effect on project success. Moreover, the effect of project duration on project success is non-linear as short and moderate levels of duration is not always certain to have significant impact on project success, but the presence of a long duration is likely to exert a positive effect on project success. Last, funds pledge have positive relationship with green crowdfunding project success, which is consistent with altruism mechanism as our study is in the context of non-profit green project [22]. 

The reminder of the study is organized as follows. An introduction to crowdfunding and a literature review of crowdfunding project success are presented, followed by the hypotheses development. The research methodology and results are discussed next. Last, implications and future research are presented.

## 2. Theoretical Background

### 2.1. Green Crowdfunding Project

Crowdfunding can help entrepreneurs, artists, and non-profits organizations to raise money from collective individuals on the Internet, because it is easier to gain financial support relative to traditional fundraising channels (e.g., venture capital, angel investment, donations) [13,23]. Therefore, more and more ventures start to launch projects on crowdfunding platforms. There are four types of business models of crowdfunding, including equity-based, lending-based, donation-based, and reward-based crowdfunding [24]. Nowadays, various types of projects are inquiring financial support through crowdfunding platforms, such as film/video, journalism, technology, public assets, and green products [6,17,19,23,24]. This study focuses on green crowdfunding projects. There are two types of green crowdfunding. One is green donation-based crowdfunding, which includes non-profit activities and aims at building a green environment by protecting the nature. The other one is green reward-based crowdfunding, which is commonly used by organizations who sell green products or services [6]. In general, green crowdfunding is a novel and popular transaction method, which can largely improve the efficiency of raising initial funds and selling innovative green products or services. 

The emergence of green crowdfunding projects, which are always launched by nonprofit organizations, is likely to be influenced by the extensive attention to environmental problems [25]. Organizations and society as a whole start to focus more on environmental protection. In a greener environment, people can live better, organizations can do business more profitably, and the society can develop better. Following the call of green and sustainable growth, environment protection becomes much more critical than before. Hence, harnessing crowdfunding, an increasing number of environment protection projects has been initiated, named as *green crowdfunding project* [6,25]. Similar to other types of donation-based projects, funders or backers of green crowdfunding projects do not necessarily receive any rewards for their donations. Moreover, it takes a long period for environment protection projects to exert effects. For example, trees can prevent sandstorm, but it takes years for trees to grow into forest. Therefore, it is also not easy for green donation-based crowdfunding projects to be successfully funded. Hence, intend to investigate how green donation-based crowdfunding projects could be successfully funded. 

### 2.2. Crowdfunding Project Success

Crowdfunding project success has received extensive attention in prior literature, especially from the perspective of fundraisers. Previous studies have explored what fundraisers can do to increase the success rate of their projects [24]. First, fundraisers can choose proper design strategies to promote project success, such as the design for the reward schema [11,12], the design of video advertisement content [10], and the design of proper funding goal and duration [20]. Second, fundraisers can carefully control the quality of their project by improving quality of project description or updating the projects intensively [9,16]. Third, fundraisers can harness their social networks to enhance project success, such as social media usage and homepage usage [9,26]. Forth, fundraisers can take advantage of social capital to help achieve project success, such as large social network size, strong social network ties, the shared meaning of project between fundraisers and funders [13,16,27]. This study focuses on design strategies that fundraisers can use to promote their environmental protection projects. 

For green donation-based crowdfunding projects, design strategies for reward schema are less applicable because these projects are charity related and thus no reward would is necessary provided at the completion of the project. However, setting proper funding goal and duration is totally under the control of fundraisers. Therefore, this study concentrates on the effects of goal setting and project duration on project success. Actually, the role of goal setting and project duration is studied directly or indirectly (i.e., being taken as control variables) by previous research [11,12,16,17,18,19,20,21,27,28]. After a scrutiny, we found that inconsistent results existed for these two as elaborated below. 

First, for the effects of goal setting, it is always justified that a higher goal would lead to lower success, and some researchers empirically verified this justification [11,12,16,17,18,20,21,27]. For example, Cordova et al. [17] found that an increase in project funding goal leaded to a lower probability and extent of success. Mollick [16] empirically reported that increasing goal size was negatively associated with success. Lagazio and Querci [20] argued that big-sized (a higher goal size) projects had a lower probability of success. Taking goal setting as a project-specific aspect, Koch et al. [21], Zhang et al. [12] and Giudici et al. [27] empirically verified that a higher goal lead to lower success. However, on the other hand, there are still several studies that have different findings. Specially, the study of Xiao et al. [11] showed that goal setting did not play a significant effect, and Li et al. [18] revealed a positive influence. 

Second, as for project duration, similar inconsistent findings were found [11,12,16,17,18,19,20,21,28]. Intuitively thinking, a longer funding duration should positively related to project success, and this is supported by some studies [17,19,20]. For example, Burtch et al. [19] proved that longer funding durations positively affected the project success in a crowd-funded marketplace for online journalism projects, because a longer duration might imply high awareness of the project and good quality of the output of the project. Cordova et al. [17] found that project duration increased the chances of success, with the consideration of the longer of fundraising closure the higher the likelihood contributions. Lagazio and Querci [20] argued that prolonged campaigns (duration) were more likely to reach the project success. That’s because potential investors need more time to fully appreciate the features of different projects to overcome information asymmetry. However, different results were also obtained. Directly or indirectly, the studies of Zhang et al. [12], Mollick [16], Clauss et al. [28], and Li et al. [18] indicated a negative relationship between project duration and project success. One reason is that duration serves as a time-control mechanism and increases the motivational impact of potential contributors. Another reason is that longer fundraising time indicate a lack of confidence [16,28]. Furthermore, taking project duration as a control variable, Xiao et al. [11] and Koch et al. [21] proposed no effects of project duration on project success. 

These studies offer various perspectives in understanding crowdfunding project success. However, there are still two limitations. First, the existing understanding of crowdfunding project success is mostly focus on the general types of projects. As green crowdfunding projects are non-profit, the determinants of project success might be different from other types of projects. The studies of green crowdfunding project success are rare. Second, inconsistent findings from previous studies indicate that the roles of goal setting and project duration might not as simple as previous studies think about. Therefore, this study intends to explore how goal setting and project duration affect project success of green crowdfunding projects by investigating their non-linear effects instead of previously studied linear ones. 

## 3. Hypotheses Development

### 3.1. Goal Setting

In the crowdfunding projects, fundraisers must set a goal in advance. Recent work on choice overload identified that goal of decision has significant influence on the robustness of choice overload [29]. Therefore, setting a proper funding goal is critical for a crowdfunding project. The goal setting and motivation theory also proposes that a goal is one of important motivations that improve performance [30]. The proper goal offers a guarantee for start and success of a project [31]. 

The existing studies indicate an inconsistent conclusion related the role of goal setting on project success. On one hand, a higher goal requires to attract more funders or backers that may hinder the success of a project [12,17,21,32]. On the other hand, a higher goal is challenging and may inspire people’s sympathy, especially for donation-based projects, because people want to help others [19,32]. This indicates that for donation-based projects a higher goal may attract more funders and then receive more funding. To the background of the above arguments and given the mixed findings on the goal setting—project success relationship referenced above, we argue that the effect of goal setting on project success is non-linear as low and moderate levels of goal setting is not always certain to have significant impact on project success, but the presence of a higher goal is likely to exert a positive effect on project success. We expect goal setting to have a non-linear effect on the project success. 

**Hypothesis 1** **(H1):**
*The crowdfunding project’s goal setting exhibits a non-linear relationship with project success.*


### 3.2. Project Duration

Duration is another item that fundraisers must set in advance when launching projects. Duration means the number of days for which a project accepts funding [16]. Setting a proper duration is also critical for fundraisers. 

A long duration can give sufficient time for projects to acquire funding, in which a project can easily receive enough funding. Deadlines serving as a time-control mechanism could help improve the effectiveness of goals and increase the motivational impact of targets based on the goal-setting theory [33]. Individuals tend to postpone the decision or action. If the duration is not long enough for individuals to make decision, they might miss the opportunity to participate in. These two mechanisms indicate that a non-linear relationship should be considered. We expect that the effect of project duration on project success is non-linear as short and moderate levels of duration is not always certain to have significant impact on project success, but the presence of a long duration is likely to exert a positive effect on project success. We thus hypothesize that: 

**Hypothesis 2** **(H2):**
*The green crowdfunding project’s’ duration setting exhibits a non-linear relationship with project success.*


### 3.3. Funds Pledge

Funds pledge is the most fundamental behavior of the crowds on the crowdfunding platforms. There are two major mechanisms utilized in the previous research of crowdfunding to explain the funds pledge behavior of the funders: herding and altruism. Herding refers to an individual’s behavior of observing or imitating others [34]. Studies on equity-based and lending-based crowdfunding platforms find evidence for this mechanism; that is, the crowds will manifest a herding behavior when they decide to pledge funds on a project [35]. Given that financial return is the major expectation in these platforms, following others is a rational way for individuals to filter out high quality projects. This implies that a more successful project in equity-based and lending-based crowdfunding platforms may receive more supports from more funders. The focus of this study is the green crowdfunding projects, in which project funders which receive intangible rewards. Hence, the funder’s pledge behavior on these platforms must be different from that of the participations on other types of crowdfunding platforms. Their goals in participating green crowdfunding platforms are helping initiators [36]. Therefore, the crowd’s participation in funds pledge should be positively related to project success. 

**Hypothesis 3** **(H3):**
*The crowd’s participations in funds pledge has positive effect on project success.*


## 4. Methodology

### 4.1. Data Collection

We collected data from Tencent Lejuan crowdfunding platform, one of the most popular donation-based crowdfunding platform in China. Tencent Lejuan crowdfunding platform enables nonprofit organizations to issue donation-based projects, such as poverty relief, disaster relief, fellowship, and environment protection, over the Internet and receive small investments from registered users in return. It is similar like CrowdRise that is a traditional donation-based crowdfunding model. In this study, we focus on green projects, and therefore we only collect data on projects related to green growth (i.e., environmental protection) from Tencent Lejuan crowdfunding platform. To ensure the sample included only successful projects, we selected projects which completed and at least one funds pelage participated in. The final project sample included 1,389 successful projects. All the data are cleaned based on the requirement (as shown in Table 1). Figure 1 illustrates the time window of a crowdfunding project. As goal setting and project duration are setting before the crowdfunding project start, it could provide support for the hypothesized causal influence of goal setting and project duration on project success. 

### 4.2. Measures

*Funds pledge*. The platform enables users to pledge in projects based on their preferences. Users can choose the amount they plan to pledge in a particular crowdfunding project. In this study, funds pledge is operationalized as the total number of funders in a particular project. 

*Crowdfunding goal setting*. Each project has a target goal that it wants to achieve. Therefore, the goal setting is operationalized as the total amount a project sets for its target. 

*Project duration*. Each project need set a time period to complete the project. Therefore, the project duration is operationalized as the total days a project last.

*Project success*. Users are able to check the status of the project from the very beginning to its completion. Because projects differ in the volume of funds that they ask for, it is improper to operationalize project success as the total amount funds. This study operationalizes project success as the completion percentage of a project, which is the ratio of the actual collected funding divide target funding goal (as shown in the formula below).
(1)$$ProjectSuccess=\frac{Acturalcollectedfunding}{Targetfundinggoal}$$

The descriptive statistics of four variables is shown in Table 1.

### 4.3. Data Analysis and Results

We used an ordinary least square (OLS) method for data analysis by using SPSS. We performed panel data random effects regression with robust standard error and pooled over ordinary least square analysis with standard error clustered by user. Clustered standard errors can control for potential heteroscedasticity.

To test our research model, we examined the research model in a regression framework as below. Following Kohtamäki et al. [37] and Fang et al. [38], we create square term (*goalsetting*^2^ and *duration*^2^) for goal setting and duration to test non-linear effect of project duration and goal setting.
(2)$$\begin{array}{cc} ProjectSuccess & =\beta_{0}+\beta_{1}*FundPledge+\beta_{2}*GoalSetting+\beta_{3}*GoalSetting^{2}\\ & +\beta_{4}*Duration+\beta_{5}*Duration^{2} \end{array}$$
where *Fundpledge* refers to number of funds pledge, *GoalSetting* refers to total amount of a project target, *GoalSetting*^2^ refers to square of number of funds goal, *Duration* refers to total number of days a project last, *Duration*^2^ refers to square of number of duration. Before conducting data analysis, Koenker statistic was used to check heteroscedasticity. The Koenker results (*x*^2^ = 10.856 with *p* = 0.212) indicated that heteroscedasticity is not exist in this study. 

Table 2 presents the results of the four studied models. The first model tests the main effects including funds pledge, goal setting, and duration. The second model enters the non-linear squared term of the goal setting. The third model adds the non-linear squared term of the duration, whereas the fourth model enters both the non-linear squared term of the goal setting and duration. We present plotted results to enable an interpretation of the marginal effects.

In model 1, the regression results suggest that funds pledge has significant effect on project success (β = 0.309, *p* < 0.01). The results show that both goal setting and duration have negative significant effect on project success (β = −0.073, *p* < 0.01; β = −0.276, *p* < 0.01).Thus H3 is supported. Model 1 explains 11.9% of the variation in project success.

Model 2 tests the non-linearity of goal setting. Model 2 confirms the positive and statistically significant non-linear effect of goal setting on project success (β = 0.401, *p* < 0.01). The positive effect of the quadratic term of goal setting indicates that the effect of goal setting on project success are not constant but rather increase at a progressive rate at higher levels of goal setting. The negative linear effect (β = −0.442, *p* < 0.01) further reveals that at low levels of a goal setting, the effect on project success is minimal. We depict this relationship graphically in Figure 2. Thus, we find support for H1.

Model 2 explains the variation of project success by 14.4%, improving model 1 where we tested the direct liner effect of project success (ΔR^2^ = 0.026). Model 3 tests the non-linearity of duration. Model 3 confirms the positive and statistically significant non-linear effect of duration on project success (β = 0.517, *p* < 0.01). The positive effect of the quadratic term of duration indicates that the effect of duration on project success are not constant but rather increase at a progressive rate at higher levels of duration. The negative linear effect (β = −0.668 *p* < 0.01) further reveals that at low levels of a duration, the effect on project success is minimal. We depict this relationship graphically in Figure 3. Thus, we find support for H2.

Model 3 explains the variation of project success by 19.9%, improving model 1 where we tested the direct liner effect of project success (ΔR^2^ = 0.08). Model 4 test the full model, including both the non-linearity of goal setting and duration. 

Model 4 confirms both the positive and statistically significant non-linear effect of duration and goal setting on project success. Model 4 explains the variation of project success by 21.4%, significantly improving model 3 where we tested the direct liner effect of project success (ΔR^2^ = 0.016).

### 4.4. Robustness Check 

To test the robustness of our model, we applied our model to another dataset from kickstarter (kickstarter.com). We collected data from kickstarter in the category of ‘Technology’, which contained 705 projects about technology crowdfunding projects. We use completed percentage of project to measure project success. The results are shown below in Table 3 and Figure 4. The results show that our results continue to hold in different platform.

### 4.5. Key Findings

This study examines the factors that affect the success of green crowdfunding project. We investigated three ways of participation antecedents, namely funds pledge, goal setting, and project duration. By collecting data from Tencent Lejuan crowdfunding platform, we have got three sets of key findings. 

First, our empirical results indicated that the crowds’ participation in funds pledge had a positive relationship with degree of project success. Previous literature proposed two underlying mechanisms to explain the crowds’ behavior of funds pledges: herding and altruism. These two mechanisms are contradictory with each other to some extent that herding plays a role for self-interests and altruism exerts an effect for others’ interests. Studies on reward-based crowdfunding revealed mixed findings that some studies indicated that funders’ participation in funds pledge will be driven by herding mechanism [18,39]; While others found the opposite results that altruism and responsibility are the key drivers [22]. Our results consistent with previous findings with Altruism mechanism. It can be explained by the research context focus on green crowdfunding project. 

Second, our results for goal setting indicated that goal setting had a non-linear relationship with the success of green crowdfunding project. Setting a proper goal is critical for crowdfunding projects. Previous studies obtained inconsistent findings related to the effects of goal setting on project success. We find that the effect of goal setting on project success is non-linear as low and moderate levels of goal setting is not always certain to have significant impact on project success, but the presence of a higher goal is likely to exert a positive effect on project success. 

Third, our empirical results indicated that the crowdfunding project duration had a non-linear relationship with degree of green project success. Previous literature showed the negative relationship between project duration and the degree of project success. Our study revealed mixed effect of project duration on project success is non-linear as short and moderate levels of duration is not always certain to have significant impact on project success, but the presence of a long duration is likely to exert a positive effect on project success.

## 5. Implications and Conclusions

### 5.1. Implications

This study has several theoretical and practical implications. First, this research contributes to the literature on crowdfunding project success. Most of the existing studies on crowdfunding project success consider general types of projects (reward-based crowdfunding) [9,10,11,12,13]. Studies on projects related to crowdfunding projects for environmental sustainability are rare. Based on our knowledge, this is one of the first studies to explore the antecedents of project success in the environment protection context. Furthermore, most of these studies only focus on whether the projects succeed or not, but not take a further step to understand the degree of project success which is also critical in influencing both success of other projects and the whole crowdfunding platform. This study takes an early attempt in this stream, and enriches the existing understanding of project success. 

Second, this study advances the understanding of the relationship between project setting (i.e., goal setting, project duration, and funders’ pledge) with the success of green crowdfunding project. Our study theorizes the relationship between project setting and project success by examining the nonlinear relationship, and proposing a non-linear relationship. This reconciles the inconsistent findings in prior research about the roles of goal setting and project duration on project success, and sets a light to further studies to examine more complex effects of project success predictors. 

Third, from a practical point of view, the nonlinear results are highly relevant for stakeholders on green crowdfunding projects. Project fundraisers can make use of the results in order to set a proper goal and duration of the green donation-based project and then to increase the chance to have a project successfully funded. Besides goal setting, founders should find a proper interval for duration of the green crowdfunding project. Furthermore, fundraisers should maximize the total funding amount within a moderate number of goal setting and duration.

Fourth, this study advances the understanding of funders’ pledge on degree of green project success. Existing studies in reward-based crowdfunding provide inconsistent results about funders’ pledge behavior. Specifically, some studies indicate that funders’ participation is driven by Herding mechanism [18,39]; that is, more successful projects attract more funders. Other studies provide opposite results that altruism is the key driver [22]; that is, less successful projects attract relatively more funders. Our study is in the context of environmental suitability, and it shows that more funders lead to more successful projects.

### 5.2. Limitations and Future Research

This study also has two limitations. First, our empirical data were crawled from a crowdfunding platform in China. Therefore, future study should attempt to replicate the results of this study by studying the Western context and compare the differences between these two contexts. 

Second, we only investigated the nonlinear effects of two project setting items (i.e., goal setting and project duration) directly on project success. Future studies should explore the conditional factors that affect green crowdfunding project success, such as cultural factors, social return etc. In addition, other project-oriented factors should also be explored about their possible linear or nonlinear effects on the success of green donation-based crowdfunding project. 

### 5.3. Conclusions

In conclusion, the current study intends to explore the non-linear relationship of project setting on the success of green donation-based crowdfunding projects. We test a non-liner decision model to include three determinants: funds pledge, goal setting, and project duration to predict project success. We crawl 1389 green crowdfunding projects from Tencent Lejuan crowdfunding platform in China. By using ordinary least square method to conduct data analysis, we find that the effect of goal setting on project success is non-linear as low and moderate levels of goal setting is not always certain to have significant impact on project success, but the presence of a higher goal is likely to exert a positive effect on project success. Results also show that the effect of project duration on project success is non-linear as short and moderate levels of duration is not always certain to have significant impact on project success, but the presence of a long duration is likely to exert a positive effect on project success. Lastly, it is revealed that funds pledge show positive relationship with green crowdfunding project success. Our work enriches the literature of crowdfunding on the project success by reconciling inconsistent findings in previous literature, and hopes to set a light for future research.

## Figures and Tables

**Figure 1 ijerph-16-00187-f001:**
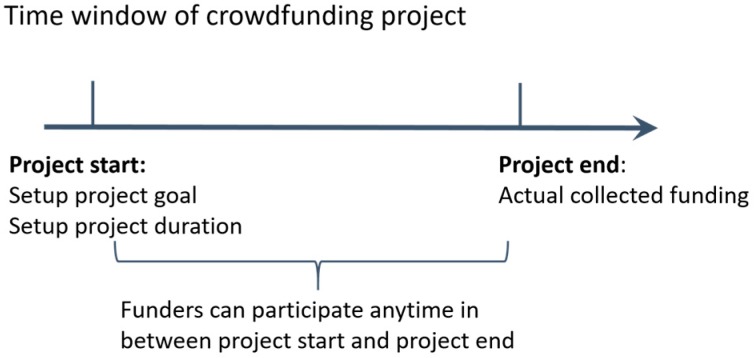
Time windows of crowdfunding project.

**Figure 2 ijerph-16-00187-f002:**
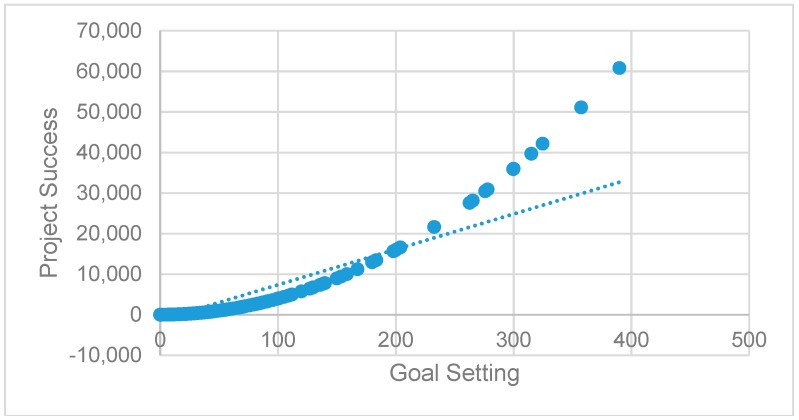
The non-linear effect of goal setting on project success.

**Figure 3 ijerph-16-00187-f003:**
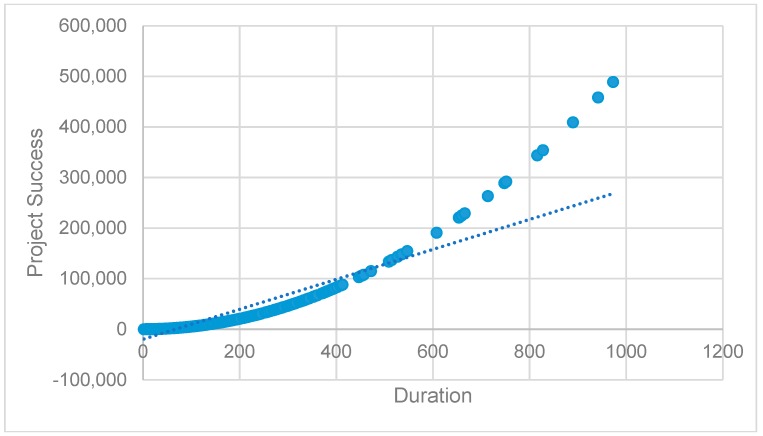
The non-linear effect of duration on project success.

**Figure 4 ijerph-16-00187-f004:**
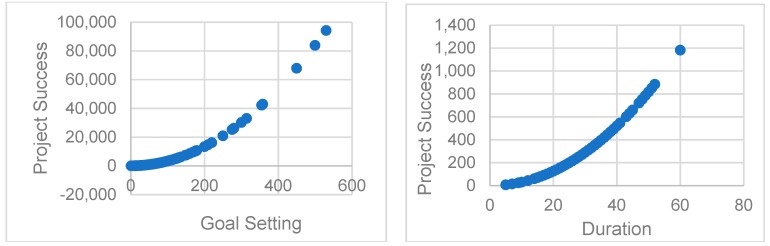
The non-linear effect of goal setting (left) and Duration (right) on project success.

**Table 1 ijerph-16-00187-t001:** Summarizes the descriptive statistics for each variables.

Variables	Min	Max	Mean	Std. Deviation
Goal setting	1000	30,000,000.000	338,089.642	1,232,957.197
Project duration	1	1940	106.12	119.954
Funds pledge	1	182,072	2,509.110	9,815.229
Project success	0.00001	2.157	0.433	0.412

**Table 2 ijerph-16-00187-t002:** Results of research model.

Variables	Model 1	Model 2	Model 3	Model 4
Funds pledge	0.309 **	0.343 **	0.213 **	0.245 **
Goal setting	−0.073 **	−0.442 **	−0.046	−0.338 **
Duration	−0.276 **	−0.272 **	−0.668 **	−0.643 **
GoalSetting^2^		0.401 **		0.315 **
Duration^2^			0.517 **	0.487 **
∆R^2^	0.121	0.026	0.080	0.016
R^2^	0.121	0.147	0.201	0.217
Adjusted R^2^	0.119	0.144	0.199	0.214
Observations	1389	1389	1389	1389

Notes: Standardized coefficients are reported; ** indicate that *p* < 0.01.

**Table 3 ijerph-16-00187-t003:** Results of robustness check.

	Full Model
Variables	
Funds pledge	0.268 **
Goal setting	−0.392 *
Duration	−0.479 **
GoalSetting^2^	0.336 *
Duration^2^	0.457 **
∆R^2^	0.016
R^2^	0.189
Adjusted R^2^	0.183
Observations	705

Notes: Standardized coefficients are reported; * indicate that *p* < 0.05, ** indicate that *p* < 0.01.

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
