# Peer review of "A Non-Liner Decision Model for Green Crowdfunding Project Success: Evidence from China"

_ijerph, 2019, doi:10.3390/ijerph16020187_

Round 1
Reviewer 1 Report
The authors took into account almost all the suggestions for completing the text. However, I still have a few cosmetic notes:
1) In the title, instead of Evidence from Environmental Sustainability Projects, I suggest Evidence from China.
2) line 32 - delete 'also'.
3) I do not think it is appropriate to combine Discussions with Conclusions. You would have to think about changing the structure of point 5. You should consider the Instructions for Authors.
4) In references, some of the journals are not in italics.
Author Response
Reviewer 1: Comments and Suggestions for Authors
1) In the title, instead of Evidence from Environmental Sustainability Projects, I suggest Evidence from China.
[R1.1] Thank you very much for your suggestion. We have changed the title to “A Non-liner Decision Model for Green Crowdfunding Project Success: Evidence from China” as suggested. |
2) line 32 - delete 'also'.
[R1.2] Thank you very much for your suggestion. There is no “also” in line 32. But we find an “also” in line 24 that should be deleted. Hence, we deleted the “also” in line 24. |
3) I do not think it is appropriate to combine Discussions with Conclusions. You would have to think about changing the structure of point 5. You should consider the Instructions for Authors.
[R1.3] Thank you very much for pointing out this issue. The section 5.1 “key findings” has been combined into section 4. In this case, section 5 includes implications, limitations, and conclusions. |
4) In references, some of the journals are not in italics.
[R1.4] Thank you very much for pointing out this issue. Following your suggestion, we have revised the references. |
Thank you very much for your constructive comments. We have revised our paper accordingly to address your concerns. We hope that you concur. |

Reviewer 2 Report
Thanks for revising the manuscript, is in much better shape. I would make additional suggestions to strengthen your arguments.
1) The emphasis of the non linearity of the model is exacerbated which may be good or bad depending on who analyses it.In my opinion from lines 134 on, a higher goal setting (quadratic term) increases the project success. That I agree, but increasing project duration (quadratic term) also leads to a higher project success. I found harder to make sense of the latter. Under which conditions it will not?
2) Why not make fund pledge quadratic as well? I want to see how it interacts with the other coefficients. Is it important if not why not?
3) In the theoretical background, two studies that contradict your findings seems to use the likelihood of project success. I understand that as a probability of completing the project. Your dependent variable use a ratio. Are these two models comparable? Yes or no and why?
4) If project success with the current ratio could be more than 1, then is already success but the variable is measuring the amount of money collected relative to a goal. I still have my doubts about it. If the ratio is within 0 and 1, the coefficients may change. Also if the model used is a probability of success, then the model may be different. If that is the case, I wonder how the new coefficients will look alike and then how the hypotheses would behave. Of course, chances are that it would be non lineal, but perhaps there would be negative signs confronting your hypotheses. Please run the other models.
5) I appreciate how you have considered new graphs and presented the results, however I do not see robustness tests. please include.
6) when you referral's to the hypotheses being supported, what does they mean? It is unclear to me as I see two variables, parameters that will support it.
Author Response
Reviewer 2: Comments and Suggestions for Authors
Thanks for revising the manuscript, is in much better shape. I would make additional suggestions to strengthen your arguments.
Thank you very much for your constructive and useful comments in your report. In this response document, we provide point-by-point responses to the issues raised by you. |
1) The emphasis of the non linearity of the model is exacerbated which may be good or bad depending on who analyses it. In my opinion from lines 134 on, a higher goal setting (quadratic term) increases the project success. That I agree, but increasing project duration (quadratic term) also leads to a higher project success. I found harder to make sense of the latter. Under which conditions it will not?
[R2.1] Thank you very much for pointing out this issue. We apologize for the confusion brought by the arguments of project duration issue. To state this issue, we add the underlying reason how duration affects the project success as follows. “a longer funding duration should positively related to project success, and this is supported by some studies [17,19,20]. For example, Burtch et al. [19] proved that longer funding durations positively affected the project success in a crowd-funded marketplace for online journalism projects, because a longer duration might imply high awareness of the project and good quality of the output of the project. Cordova et al. [17] found that project duration increased the chances of success, with the consideration of the longer of fundraising closure the higher the likelihood contributions. Lagazio and Querci [20] argued that prolonged campaigns (duration) were more likely to reach the project success. That’s because potential investors need more time to fully appreciate the features of different projects to overcome information asymmetry. However, different results were also obtained. Directly or indirectly, the studies of Zhang et al. [12], Mollick [16], Clauss et al. [28], and Li et al. [18] indicated a negative relationship between project duration and project success. One reason is that duration serves as a time-control mechanism and increases the motivational impact of potential contributors. Another reason is that longer fundraising time indicate a lack of confidence [16,28].” |
2) Why not make fund pledge quadratic as well? I want to see how it interacts with the other coefficients. Is it important if not why not?
[R2.2] Thank you very much for pointing out this issue. The focus of this study is the green crowdfunding projects, in which project funders which receive intangible rewards. We argue that the crowds’ participation in funds pledge is positively associated with project success. Hence, we didn’t make fund pledge quadratic. |
3) In the theoretical background, two studies that contradict your findings seems to use the likelihood of project success. I understand that as a probability of completing the project. Your dependent variable use a ratio. Are these two models comparable? Yes or no and why?
[R2.3] Thank you very much for pointing out this issue. In the context of green crowdfunding, if the project can’t reach the goal, the project continue to run based on the percentage of success. That’s the reason we use a ratio as the dependent variable. This “take-it-all” fundraising model is different from “all-or-nothing” model. The different finds between the current study and previous study is our research focus. We argue different effects and use different research context. |
4) If project success with the current ratio could be more than 1, then is already success but the variable is measuring the amount of money collected relative to a goal. I still have my doubts about it. If the ratio is within 0 and 1, the coefficients may change. Also if the model used is a probability of success, then the model may be different. If that is the case, I wonder how the new coefficients will look alike and then how the hypotheses would behave. Of course, chances are that it would be non lineal, but perhaps there would be negative signs confronting your hypotheses. Please run the other models.
[R2.4] Thank you for pointing this issue. In Tencent Lejuan crowdfunding platform, if the project can’t reach the goal, the project continue to run based on the percentage of success. This is different from “all-or-nothing” fundraising model (i.e., a fundraiser sets a funding goal in advance, and receives no funds if this goal cannot be reached). Therefore we use the ratio to measure the degree of project success. |
5) I appreciate how you have considered new graphs and presented the results, however I do not see robustness tests. please include.
[R2.5] Thank you for pointing the issue. We have added robustness check in section 4.4. “To test the robustness of our model, we applied our model to another dataset from kickstarter (kickstarter.com). We collected data from kickstarter in the category of ‘Technology’, which contained 705 projects about technology crowdfunding projects. We use completed percentage of project to measure project success. The results are shown below in Table 2 and Figure 3. The results show that our results continue to hold in different platform.” |
6) when you referral's to the hypotheses being supported, what does they mean? It is unclear to me as I see two variables, parameters that will support it.
[R2.6] Thanks for pointing this issue. Sorry for confusing you in Table 3. As we have indicated hypotheses support in results, we delete table 3 in this verson. Thank you very much for your constructive comments. We have revised our paper accordingly to address your concerns. We hope that you concur. |

This manuscript is a resubmission of an earlier submission. The following is a list of the peer review reports and author responses from that submission.
Round 1
Reviewer 1 Report
Issues taken up by the authors have a significant socio-economic importance, but the text needs some adjustments and additional information:
1. Considering the main purpose of the paper, I would suggest changing the title to include a category of green crowdfunding.
2. Not sufficient abstract. Please read, how to prepare a strong abstract:
https://www.aje.com/en/arc/make-great-first-impression-6-tips-writing-strong-abstract/
3. The main thesis is rather trivial and obvious: line 42-44.
4. Literature review is incomplete. There is no reference to many important items, e.g.:
http://ira.lib.polyu.edu.hk/bitstream/10397/44202/1/Lam_Crowdfunding_Renewable_Sustainable.pdf
http://www.crowdfundres.eu/wp-content/uploads/2017/11/Crowdfunding-for-green-projects-in-Europe-2017.pdf
https://cordis.europa.eu/news/rcn/142115_en.html
https://www.researchgate.net/publication/325756448_FINANCING_GREEN_PROJECTS_FROM_GREEN_CROWDFUNDING_TO_ENVIRONMENTAL_IMPACT_BONDS
https://www.researchgate.net/publication/303415405_Chapter_21_The_Crowdfunding_of_Renewable_Energy_Projects
5. The conclusions should be supported to a greater extent by the results. Take a look, how to write a conclusion for a research paper:
https://www.wikihow.com/Write-a-Conclusion-for-a-Research-Paper
Reviewer 2 Report
Generally, the text lacks of references, expecially of high impact factor journals. Therefore, the scientific soundenss of the paper is very poor, as well as the literature support.
It is not clear to me the novelty of your paper. Why is it new in comparison with previous literature?
Morevoer, in China, are there problems about foundrasing? What is the issue that you would like to solve? You did not explain the structure of the projects, neither the main issue find out during the years. No data available in the paper about the current management in China. No case study discussed.
The conclusions of your reserach are too poor for a scientific paper. What is the main impact of the research? What is the novelty? For whom is it useful? What is the target of your research and paper?
In my opinion, your article is not suited for a scientific journal.
Specific comments.
Abstract:
Results too general, as well as the method used.
What is the novelty of your study?
Why is it useful for the scientific audience?
Introduction: Introduction too long, without exploring case studies or explaining the main issue, or the topic of the paper and the novelty of your Research. It lacks references and scientific support.
L23 I suggest reformulate the sentence.
L23-31 In my opinion you should restructure the paragraph. It could open to missunderstandings
L39 references. Please, add information about the platform
L42-43 With which scientific basis?
L47 No examples, no references, no scientific support. The introduction should be rewritten
L51 et al., not etc.
L 60 references. Only personal opinions, with no scientific support.
L63 too general. The references should be reported separately, explaining where they apply it, how, why, what are the main issues, why your research is required, who used the same method that you suggested...
L 65 why non linear? Why this method? It is not clear the novelty of your research and the aid that it could provide to the readers of the journal.
L72 The research area is very limited. You should better explain the context, the method, the issues identified, the areas of interest and the main goals.
L74 Previous leterature: only one paper???
L77 Is it a result??? Why did you introduce in the introduction section?
L80 Not clear the concept of "previous literature" since you cited only a paper
L82-85 Not clear
Methods - Results - Conclusions
L94 references
L98 references
L105 These years? What do you mean? Too poor scientific support
L111 Is it at international level? Or local?
L114 too general, lack of references
L116 With which kind of bases you state it?
L122 You should be more precise. So, what is the novelty of your research?
L124 what do you mean?
L 135 please, can you provide examples? Again, too general
L136 In what sense? It seems a contraddiction if reported in this way
L 148 it is a repetition
L154 what do you mean?
L153 same references used before. Again, lack of references within the text.
L157 With what bases do you introduce this hypothesis? Is it in general or only within your country?
L158 what do you mean with U-shaped???? You did not introduce the concept of U-shape relationship
L171 Not clear. I think that you mean that there is a more apporpriate time for collecting funds for the projects. However, it is function of various factors: income of the population, areas of interest (international, national, regional, municipal), the dimension of the project, the amount required, the subject of the prroject.... You never speak about it within your paper.
L186 This statement is not clear. How can be related with success? Do you mean more funds collection?
You never provide tangible examples of successful projects, neither of negative...
L190 what about the others?
L194 too general. No references are provided. So your study is limited to China and to this platform?
L195 No data about these projects.
L197 why only successful? So, what would you like to solve? How many project, in percentage, are not successful? Why? What is the statement of the problem?
L197 reference time?
L198 Table 1 is not clear
L 200 I suggest to be coherent with the ones listed in Table 1
L202 It is not clear the unit of measure. What about the minimum (equal to 0)? Data are only about the platform introduced in section 4.1? The study is too limited and not useful for the international stakeholdres...
L206 minumum is equal to zero? How could it be possible???
L210 an the minimum is zero?
L211 Below? Here there is only Table 1. Please, refere to the equation number
L214 chapter 5 - results
L221 add explanations about coefficient beta. These are results, not methods!
L236 Results too poor of contents. You only provided the statistical analysis introduced by a the software SPSS. The argument introduced within the introduction and method section is not scientifically supported, neither the variables used for the study. It is not clear the method for assessing the crowdfunding of the project, neither the problem which should be solved thank to your study.
L244 It is the introduction of an abstract or conclusion section.
L247 too poor literature review.
L261 It is not clear to me which indication you provide to the stakeholders. So, what do they do for improving crowed funding? In my opinion, your study has no impact in this topic.
L267 your study is too limited in the study area, while there is not description about the time. How can you compare your study with others? You never explain the specific charachteristic of other researches
L271 Do you explain it? Where?
L277 no graphical explanation? You statistically prove it, however, without any support. Data are not reported, neither explanation about the results in the context, explaining how your hypothesis could be considered for all the "green" projects...
L282 How????
L284 what is the propoer interval? no answer!
L290 repetition
L294 This is an obvious result...
L296 This is a big limitation. The paper is not suited for publication in the current form.
Reviewer 3 Report
The paper aims to show that environmental development project success follows a non lineal decisión making process.
I appreciate the intention by the authors to try to explain how crowdfunding decisión making Works, unfortunately I think the article is misguiding starting from the title. I pressume that the fact that the model specification is non lineal is sufficient for the authors to imply that all decisions are non lineal, and while I understand that it implies causality and the regression model as is presents correlations not causality.
The dependent variable "Project success" is supposed to be the percentage completion of the project and yet in the descriptive stats have a max value of 2.157 implying that the completion success is way completed. Please explain.
Now, another concer is the U shape decisión making in which authors make most of their arguments.
1) According to the number provided by the authors and the results from the regression analysis, I do not believe there is a U shape process. In both cases it is an exponential growth, I would expect that in fact as the project duration extends, the success decreases suggesting an inverted U shape as many variables behave overtime in particular with the functional form suggested. In line 168 the authors highlined the opposite to their findings.
2) similarly occurs with Goal setting. Please see PDF of Excel spreadsheet attached in which I use your numbers to show the inconsistencies.
3) Given that Funds pledge has a positive coefficient, then there is no way that Project success will take the values presented unless something is wrong in the model that is not presented by the authors.
3) The Results in row 237 page 7 are not presented as the expected
In row 230-232 you present the information on the coefficients without clarify the index
Several grammatuical errors
I want to think that I do not understand the statistics behind your model and I will gladly re-read your article . Best of lucks
